# INVARIANCE AND INVERSE STABILITY UNDER ReLU

## ABSTRACT

We flip the usual approach to study invariance and robustness of neural networks by considering the non-uniqueness and instability of the inverse mapping. We provide theoretical and numerical results on the inverse of ReLU-layers. First, we derive a necessary and sufficient condition on the existence of invariance that provides a geometric interpretation. Next, we move to robustness via analyzing local effects on the inverse. To conclude, we show how this reverse point of view not only provides insights into key effects, but also enables to view adversarial examples from different perspectives.

## 1 INTRODUCTION

Invariance and stability/robustness are two of the most important properties characterizing the behavior of a neural network. Due to growing requirements like robustness to adversarial examples (Szegedy et al., 2014) and the increasing use of deep learning in safety-critical applications, there has been a surge in interest in these properties. Invariance and stability are considered to be the key mechanisms in dealing with uninformative properties of the input (Achille & Soatto, 2017; Mallat, 2016) and are studied from the information theoretical perspective in form of the loss of information about the input (Tishby & Zaslavsky, 2015; Saxe et al., 2018).

Invariance and stability are also tightly linked to robustness against adversarial attacks (Cisse et al., 2017; Tsuzuku et al., 2018; Simon-Gabriel et al., 2018), generalization (Sokolić et al., 2017; Gouk et al., 2018) and even the training of Generative Adversarial Networks (Miyato et al., 2018). In general, stability is studied via two basic properties: 1) locally via a norm of the Jacobian (Sokolić et al., 2017; Simon-Gabriel et al., 2018), 2) globally via the Lipschitz constant (Cisse et al., 2017; Miyato et al., 2018; Tsuzuku et al., 2018). From a high-level perspective, both of these approaches study an upper bound on stability as the Lipschitz constant and a Jacobian norm quantifies the highest possible change under a perturbation with a given magnitude. We, unlike the approaches above, aim to broaden our understanding by analyzing the lowest possible change under a perturbation.

More formally, we study which perturbations $\Delta x$ do not (or only little) affect the outcome of a network $F$. Our analysis considers a given input data point $x$ and investigates the $\Delta x$'s, such that

$$F(x) = F(x + \Delta x) \quad \text{(invariant)} \qquad \text{or} \qquad \|F(x) - F(x + \Delta x)\| \leq \varepsilon \quad \text{(stable)},$$

where a small $\varepsilon > 0$ is given. While these properties can be crucial for many discriminative tasks (Mallat, 2016), the model could be flawed if perturbations that alter the semantics have only a minor impact on the features. This is a reverse perspective on adversarial examples (Szegedy et al., 2014), which commonly considers small input perturbations that lead to large changes and thus to arbitrary decisions of the network.

This flipped view and the study of smallest changes calls for a different approach: we study the instabilities of the inverse instead of the stabilities of the forward mapping. In particular, if $F$ is invariant to perturbations $\Delta x$, then $x$ and $x + \Delta x$ lie in the preimage of the output $z = F(x)$, i.e. $F$ is not uniquely invertible. Robustness towards large perturbations induces an instable inverse mapping as small changes in the output can be due to large changes in the input.

Based on the piecewise linear nature of ReLU networks (Montufar et al., 2014), we characterize the preimage of ReLU-activations as a single point, finite (bounded) or infinite (unbounded). Further, we study the stability of the linearization of rectifier networks via its singular values. To illustrate these locally changing properties and to demonstrate their tight connection, we visualize the behavior on a synthetic problem in Figure 1.

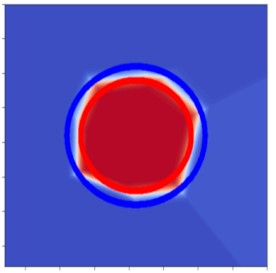 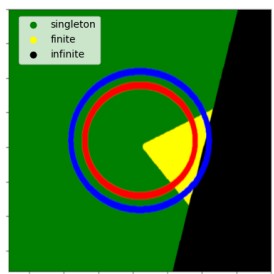 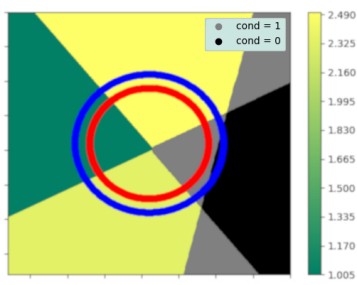

Figure 1: Left: Prediction of a small ReLU-network (one hidden layer with 3 neurons) trained to distinguish samples from two circles. Middle: Characterization of the preimage of first layer activations into unbounded (infinite), compact (finite) or unique (a single point). Right: Condition of the linearization of the first layer at each point.
As ReLU-layers are piecewise linear, the local behavior is constant on polytopes. Further, the regions with infinite/finite preimages correspond to regions with condition number of one or zero, while singleton preimages link to condition numbers larger than one. Thus, both properties are tightly connected and investigating one property alone yields an incomplete picture.

Our contributions are as follows:

- We derive conditions when the preimage of an output of a ReLU-layer has finite or infinite volume or is a single point. Based on these conditions, we derive an algorithm to check these conditions and exemplify its usability by applying it to investigate the preimages of a trained network. (See Section 2.)

- We study the stability of the inverse via analyzing the linearization at a point in input space, which is accurate within a polytope. We provide upper bounds on the smallest singular value of a linearization and prove how the removal of uncorrelated features could effect the stability of the inverse mapping. Based on these ideas, we experimentally demonstrate how singular values evolve over the different layers in rectifier networks. (See Section 3.)

- We introduce a reverse view on adversarial examples and connect it to invariance and robustness by leveraging our analysis of preimages. (see Section 5)

## 1.1 RELATED WORK

While analyzing invariance and robustness properties is a major topic in theoretical treatments of deep networks (Mallat, 2016), studying it via the inverse is less common. Several works like Mahendran & Vedaldi (2015), Mahendran & Vedaldi (2016) or Dosovitskiy & Brox (2016) focus on reconstructing inputs from features of convolutional neural networks (CNNs) to visualize the information content of features. Instead, we investigate potential mechanisms affecting the invertibility.

Carlsson et al. (2017) gives a first geometrical view on the shape of preimages of outputs from ReLU layers, which is directly related to the question of injectivity of the mapping under ReLU. Shang et al. (2016) analyzes the reconstruction property of cReLU (concatenated ReLU); however, the more general situation of using the standard rectifier is not studied. A notable other line of work assumes random weights in order to derive guarantees for invertibility, see Gilbert et al. (2017) or Arora et al. (2015), whereas we focus on the preimage of ReLU-activations without assumptions on the weights.

Moreover, several reversible network structures were recently proposed (Gomez et al., 2017; Chang et al., 2018; Jacobsen et al., 2018). Most notably, in Jacobsen et al. (2018) a bijective network, up to its last layer, was trained successfully on ImageNet which does not exhibit any invariance. However, the network is very robust towards many directions in the input which is reflected in a strongly instable inverse. Hence, even carefully designed network show at least one of the two effects (invariance and robustness) studied in this work. Especially stability has seen growing interest due to adversarial examples (Szegedy et al., 2014), yet stability is mostly studied with respect to the forward mapping, see e.g. Cisse et al. (2017).

Two main resources for our view of rectifier networks as piecewise linear models are Montufar et al. (2014) and Raghu et al. (2017). Closest to our approach is the work of Bruna et al. (2014) on global statements of injectivity and stability of a single layer including ReLU and pooling. The authors focus on global injectivity and stability bounds via combinatorial statements over all configurations attainable by ReLU and pooling. These conditions are valid on the entire input space, while the restriction to parts of the input space may be far from these worst-case conditions.

Further works focus on applications like inverse problems with learned forward models (Jensen et al., 1999; Lu et al., 1999) and parameter estimation problems (Lähivaara et al., 2018), which are often formulated as inverse problems and require the inversion of networks.

## 1.2 NOTATION

In this section, we briefly state our notation as a reference:

- Input: $x = x^0 \in \mathbb{R}^{d_0}$, sometimes shortened to $d := d_0$.
- Pre-activations: $z^l = A^l x^{l-1} + b^l \in \mathbb{R}^{d_l}$, with weight matrix $A^l \in \mathbb{R}^{d_l \times d_{l-1}}$ and bias $b^l \in \mathbb{R}^{d_l}$.

- Activation: $x^l = \phi(z^l) \in \mathbb{R}^{d_l}$, where $\phi : \mathbb{R} \to \mathbb{R}$ the pointwise applied activation function, if not specified differently $g :=$ ReLU.
- Number of layers: $L \in \mathbb{N}$

- Entire network: $F : \mathbb{R}^d \ni x \mapsto F(x) := z := z^L \in \mathbb{R}^{d_L}$, sometimes short $D := d_L$.

For matrices $A \in \mathbb{R}^{m \times n}$ and $I \subset [m] := \{1, \ldots, m\}$, $A|_I$ denotes the matrix consisting of the rows of $A$ whose index is in set $I$ – analogously for vectors. Also $A|_{y \succ 0}$ describes the restriction to the index set $\{i : y_i > 0\}$ for $y \in \mathbb{R}^m$, analogously for $\prec, =, \preceq, \succeq$. For vectors $y \in \mathbb{R}^m$, $y \succ 0$ is the elementwise relation, analogously for $\prec, =, \preceq, \succeq$. Furthermore, we define $\mathcal{N}(A)$ as the null space of a matrix $A$. The Euclidean inner product is denoted by $\langle \cdot, \cdot \rangle$.

For every matrix $A \in \mathbb{R}^{m \times n}$ with the rows $a_i$, $i \in [m]$, we associate the set $A = \{a_i\}_{i=1}^m$. Vice versa, we associate every finite set in $\mathbb{R}^n$ with a matrix (only possible up to permutation of the indices).

## 2 PREIMAGES OF ReLU LAYER

### 2.1 THEORETICAL ANALYSIS

In this section, we analyze different kinds of preimages of a ReLU-layer and investigate under which conditions the inverse image of a given point is a singleton (a set containing exactly one element) or has finite/infinite volume. These conditions will yield a simple algorithm able to distinguish between these different preimages, which is applied in Section 2.2.

For the analysis of preimages of a given output one can study single layers separately or multiple layers at once. However, since the concatenation of two injective functions is again injective while a non-injective function followed by an injective function is non-injective, studying single layers is crucial. We therefore develop a theory for the case of single layers in this section. Notice that in case of multiple layers one is also required to investigate the image space of the previous layer.

We will focus our study on the most common activation function, ReLU. One of its key features is the non-injectivity, caused by the constant mapping on the negative half space. It provides neural networks with an efficient way to deploy invariances. Basically all other common activation functions are injective, which would lead to a straightforward analysis of the preimages. However, injective activations like ELU (Clevert et al., 2016) and Leaky ReLU (Maas et al., 2013) only swap the invariance for robustness, which in turn leads to the problem of having instable inverses. This question of stability will be analyzed in more detail in Section 3.

We start by introducing one of our main tools – namely the *omnidirectionality*.

**Definition 1 (Omnidirectionality)**

*i)* $A \in \mathbb{R}^{m \times n}$ *is called omnidirectional if every linear open halfspace in $\mathbb{R}^n$ contains a row of $A$, i.e. for every given $x \in \mathbb{R}^n \setminus \{0\}$ there exists an index $i \in [m]$, such that $\langle a_i, x \rangle > 0$.*

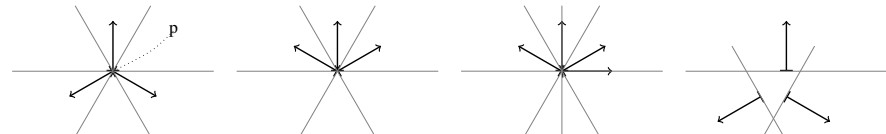

Figure 2: Gray lines are hyperplanes with normal vectors (arrows) from the rows of $A$ and translation $b$. Left: Omnidirectional tuple $(A, b)$ for $p \in \mathbb{R}^2$, as hyperplanes intersect in $p$ and normal vectors are omnidirectional. Two in the middle: Intersection in $p$, but vector-free halfspaces (hence, not omnidirectional). Right: hyperplanes do not intersect in a point, but normal vectors are omnidirectional.

**ii)** $A \in \mathbb{R}^{m \times n}$ and $b \in \mathbb{R}^m$ are called omnidirectional for the point $p \in \mathbb{R}^n$ if $A$ is omnidirectional and $b = -Ap$.

Thus, if $A$ is omnidirectional, for every direction of a hyperplane through the origin forming two halfspaces, there is a vector from the rows of $A$ inside each open halfspace, hence the term *omnidirectional* (see Figure 2 for an illustration). Note that the hyperplanes are due to ReLU as it maps the open halfspace to positive values and the closed halfspace to zero. A straightforward way to construct an omnidirectional matrix is by taking a matrix whose rows form a spanning set $\mathcal{F}$ and use the vertical concatenation of $\mathcal{F}$ and $-\mathcal{F}$. This idea is related to cReLU (Shang et al., 2016).

The following Corollary gives several equivalent formulations of omnidirectionality, which will turn out to be useful for the proof of the subsequent Theorem 4 in this section. The short proofs of the statements are provided in Appendix A1.

**Corollary 2 (Equivalences of omnidirectionality)** *The following statements are equivalent:*

**i)** $A \in \mathbb{R}^{m \times n}$ *is omnidirectional.*

**ii)** $Ax \preceq 0$ *implies* $x = 0$, *where* $x \in \mathbb{R}^n$.

**iii)** *There exists a unique* $x \in \mathbb{R}^n$, *such that* $Ax \preceq 0$.

**iv)** *There exist no* $x \in \mathbb{R}^n \setminus \{0\}$, *such that* $Ax \preceq 0$.

More importantly, omnidirectionality is directly related to the ReLU-layer preimages and will provide us with a method to characterize their volume (see Theorem 4). To analyze such inverse images, we consider $y = \text{ReLU}(Ax + b)$ for a given output $y \in \mathbb{R}^m$ with $A \in \mathbb{R}^{m \times n}$, $b \in \mathbb{R}^m$ and $x \in \mathbb{R}^n$. If we know $A, b$ and $y$, we can write the equation as the following mixed linear system:

$$A|_{y \succ 0} x + b|_{y \succ 0} = y|_{y \succ 0} \tag{1}$$
$$A|_{y = 0} x + b|_{y = 0} \preceq 0, \tag{2}$$

where $A|_{y \succ 0}$ denotes the restriction of the matrix $A$ to the rows, which are specified by the index set $\{i : y_i > 0\}$ (see Section 1.2 for the used notation).

**Remark 3** *It is possible to enrich the mixed system to include conditions/priors on $x$ (e.g. $x \in \mathbb{R}^n_{\geq 0}$).*

The inequality system in equation 2 links its set of solutions and therefore the volume of the preimages of the ReLU-layer with the omnidirectionality of $A$ and $b$. Defining $\overline{A} := AO^T$, where $O \in \mathbb{R}^{k \times n}$ denotes an orthonormal basis of $\mathcal{N}(A|_{y \succ 0})$ with $k := \dim \mathcal{N}(A|_{y \succ 0})$ and $\overline{b} := b|_{y \preceq 0} + A|_{y \preceq 0}(P_{\mathcal{N}(A|_{y \succ 0})^{\perp}} x)$, where $P_V$ denotes the orthogonal projection into the closed space $V$, leads to the following main theorem of this section, which is proven in Appendix A1.

**Theorem 4 (Preimages of ReLU-layers)** *Let* $\overline{A}, \overline{b}$ *and* $k = \dim \mathcal{N}(A|_{y \succ 0})$ *be as above. The preimage of a point $y$ under a* ReLU-*layer is*

**i)** *for $k = 0$ a singleton.*

**ii)** *for $k > 0$ a singleton, if and only if there exists an index set $I$ for the rows of $\overline{A}$ and $\overline{b}$, such that $(\overline{A}|_I, \overline{b}|_I)$ is omnidirectional for some point $p \in \mathbb{R}^k$.*

***iii)*** *for $k > 0$ a compact polytope with finite volume, if and only if $\overline{A}$ is omnidirectional.*

Thus, omnidirectionality allows in theory to distinguish whether the inverse image of a ReLU-layer is a singleton, a compact polytope or has infinite volume. However, obtaining a method to check if a given matrix is omnidirectional is crucial for later numerical investigations. For this reason, we will go back to the geometrical perspective of omnidirectionality (see Figure 2). This will also help us to get a better intuition on the frequency of occurrence of the different preimages. The following Theorem 5 gives another geometrical interpretation of omnidirectionality, whose short proof is given in Appendix A1.

**Theorem 5 (Convex hull)** *A matrix $A \in \mathbb{R}^{m \times n}$ is omnidirectional if and only if $0 \in \mathrm{Conv}(A)^{\mathrm{o}}$, where $\mathrm{Conv}(A)^{\mathrm{o}}$ is the interior of the convex hull spanned by the rows of $A$ (see Definition 10 in Appendix A1).*

Therefore, the matrix must contain a simplex in order to be omnidirectional, as the convex hull of the matrix $A \in \mathbb{R}^{m \times n}$ has to have an interior. Hence, we have the following:

**Corollary 6** *If $A \in \mathbb{R}^{m \times n}$ is omnidirectional, then $m > n$.*

By considering the geometric perspective, a tuple $(A \in \mathbb{R}^{m \times n}, b \in \mathbb{R}^m)$ is omnidirectional for a point $p \in \mathbb{R}^n$, if and only if the $m$ hyperplanes generated by the rows of $A$ with bias $b$ intersect at $p$ and their normal vectors (rows of $A$) form an omnidirectional set. We can use Corollary 6 to conclude that singleton preimages of ReLU-layers are very unlikely to happen in practice (if we do not design for it), since a necessary condition is that $n + 1$ hyperplanes have to intersect in one point in $\mathbb{R}^n$. Therefore we conclude, that singleton preimages of ReLU layers in practice only and exclusively occurs, if the mixed linear system already has sufficient linear equalities.

**Algorithm to check uniqueness:** The above results can be used to derive an algorithm to check whether a preimage of a given output is finite, infinite or just a singleton. A singleton inverse image is obtained as long as $\mathrm{rank}(A|_{y \succ 0}) = n$ holds true, which can be easily computed. To distinguish preimages with finite and infinite volumes, it is enough to check if $\overline{A}$ is omnidirectional (see Theorem 4iii), which can be done numerically by using the definition of the convex hull, Theorem 5 and Corollary 6. This leads to a *linear programming problem*, which is presented in Appendix A3 and was also used to create Figure 1.

## 2.2 NUMERICAL ANALYSIS

In this section, we demonstrate for a simple model that the preimage of a layer can be a singleton, infinite or finite depending on the given point. For this purpose, we trained a MLP with two hidden ReLU layers of size 3500 and 784 on MNIST (LeCun & Cortes, 2010). We chose the layer size of 3500, because the likelihood of having roughly 784 (input dimension of MNIST) positive outputs was high for this setting. In Figure 4, we plotted the number of samples in the test set that have infinite (red curve) or finite (blue curve) preimages over the number of positive outputs. It can be assumed that all samples which have more or equal to 784 (the input dimension) positive outputs have a singleton preimage and are therefore finite. In the dark gray region between 723 and 784, both effects occurred, which can be seen by the overlap of the red and blue curve.
To determine whether a preimage for less than 784 positive outputs was compact we used Theorem 4iii and the algorithm described in Appendix A3.

## 3 STABILITY

### 3.1 THEORETICAL ANALYSIS

In this section we analyze the robustness of rectifier MLPs against large perturbations via studying the stability of the inverse mapping. Concretely, we study the effect of ReLU on the singular values of the linearization of network $F$. While the linearization of a network $F$ at some point $x$ only provides a first impression on its global stability properties, the linearization of ReLU networks is exact in some neighborhood due to its piecewise-linear nature (Raghu et al., 2017). In particular, the

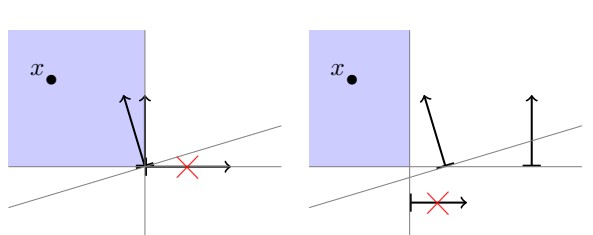

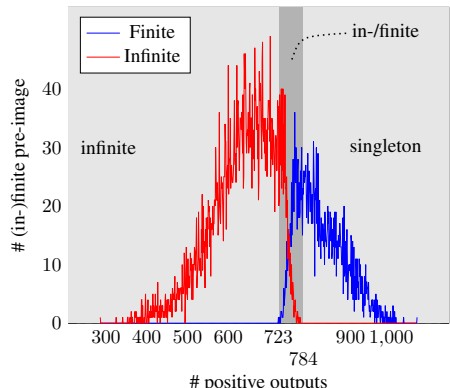

Figure 3: Removal of vectors due to ReLU (red crosses) for the marked points $x$ (left:unbiased setting, right: biased setting). The remaining vectors are only weakly correlated to the removed one, thus yielding an unstable inverse.

Figure 4: The number of (in-)finite volumed preimages of a ReLU layer over the test set of MNIST. Only within the gray strip we see finitely and infinitely volumed preimages.

input space $\mathbb{R}^d$ of a rectifier network $F$ is partitioned into convex polytopes $P_F$, corresponding to a different linear function on each region (see Figure 1). Hence, for each polytope $P$ in the set of all input polytopes $P_F$, the network $F$ can be simplified as $F(x) = A_P x + b_P$ for all $x \in P$.

In particular, each of the linearized matrices $A_P$ can be written via a chain of weight matrix multiplications that incorporates the effect of ReLU. To this end, the following definition introduces admissible index sets that formalize all possible local behaviors (Bruna et al., 2014) and diagonal matrices to locally model the effect of ReLU, see Wang et al. (2016):

**Definition 7 (Admissible index sets, ReLU as diagonal matrix)** *An index set $I^l$ for a layer $l$ is admissible if*

$$\bigcap_{i \notin I^l} \{x^l : \langle x^l, a_i^l \rangle > -b_i\} \cap \bigcap_{i \in I^l} \{x^l : \langle x^l, a_i^l \rangle \leq -b_i\} \neq \emptyset.$$

*Further, let $D_I$ denote a diagonal matrix with $(D_I)_{ii} = 1$ for $i \notin I$ and $(D_I)_{ii} = 0$ for $i \in I$, where $I$ is an admissible index set. Using this notation, the mapping of pre-activation $z \in \mathbb{R}^d$ under ReLU can be written as*

$$\text{ReLU}(z) = D_I z \quad \text{with } I = \{i \in [d] : z_i \leq 0\}.$$

Thus, the linearization $A_P$ of a network with $L$ layers is a matrix chain $A_P = A^L D_{I^{L-1}} A^{L-1} \cdots D_{I^1} A^1$, where $A^l$ are the weight matrices of layer $l$ and $I^l := \{i \in [d_l] : (A^l x^{l-1} + b^l)_i \leq 0\}$.

Of special interest for a stability analysis is the range of possible effects by the application of the rectifier. Since the effect by ReLU corresponds to the application of $D_I$ for admissible $I$, we now turn to studying the changes of the singular values of a general matrix $A$ compared to $D_I A$. For example, the matrix $A$ could represent the chain of matrix products up to pre-activations in layer $l$. Then, the effect of ReLU can be globally upper bounded:

**Lemma 8 (Global upper bound for largest and smallest singular value)** *Let $\sigma_l$ be the singular values of $D_I A$. Then for all admissible index sets $I$, the smallest non-zero singular value is upper bounded by $\min\{\sigma_l : \sigma_l > 0\} \leq \tilde{\sigma}_k$, where $k = N - |I|$ and $\tilde{\sigma}_1 \geq ... \geq \tilde{\sigma}_N > 0$ are the non-zero singular values of $A$.*
*Furthermore, the largest singular value is upper bounded by $\max\{\sigma_l : \sigma_l > 0\} \leq \tilde{\sigma}_1$.*

Lemma 8 analyzes the best case scenario with respect to the highest value of the smallest singular value. While this would yield a more stable inverse mapping, one needs to keep in mind that $\mathcal{N}(A_P)$ grows by the corresponding elimination of rows via $D_I$. Moreover, reaching this bound is very

unlikely as it requires the singular vectors to perfectly align with the directions that collapse due to $D_i$. Thus, we now turn to study effects which could happen locally for some input polytopes $P$.

An example of a drastic effect through the application of ReLU is depicted in Figure 3. Since one vector is only weakly correlated to the removed vector and the situation is overdetermined, removing this feature for some inputs $x$ in the blue area leaves over the strongly correlated features. While the two singular values of the 3-vectors-system were close to one, the singular vectors after the removal by ReLU are badly ill-conditioned. As many modern deep networks increase the dimension in the first layers, redundant situations as in Figure 3 are common, which are inherently vulnerable to such phenomena. For example, Rodríguez et al. (2017) proposes a regularizer to avoid such strongly correlated features. The following lemma formalizes the situation exemplified before:

**Lemma 9 (Removal of weakly correlated rows)** *Let $A \in \mathbb{R}^{m \times n}$ with rows $a_j$ and $I \subseteq [m]$. For a fixed $k \in I$ let $a_k \in \mathcal{N}(D_I A)^\perp$. Moreover, let*

$$\forall j \notin I : |\langle a_j, a_k \rangle| \leq c \frac{\|a_k\|_2}{\sqrt{M}}, \tag{3}$$

*with $M = m - |I|$ and constant $c > 0$. Then for the singular values $\sigma_l \neq 0$ of $D_I A$ it holds*

$$0 < \sigma_K = \min\{\sigma_l : \sigma_l \neq 0\} \leq c.$$

Note that $I$ has to be admissible when considering the effect of ReLU.
Lemma 9 provides an upper bound on the smallest singular value, given a condition on the correlation of all $a_j$ and $a_k$. However, the condition 3 depends on the number $M$ of remaining rows $a_j$. Hence, in a highly redundant setting even after removal by ReLU (large $N$), $c$ needs to be large such that the correlation fulfills the condition. Yet, in this case the upper bound on the smallest singular value, given by $c$, is high. We discuss this effect further and provide quantitative results in the Appendix A5.

**Effect under multiple layers:** For the effect of ReLU applied to multiple layers, we are particularly interested in following questions:

- Can the application of another layer have a pre-conditioning effect yielding a stable inverse?
- What happens when we only compose orthogonal matrices which have stable inverses?

Note that a way to enforce an approximate orthogonality constraint was proposed for CNNs in Cisse et al. (2017), however only for the filters of the convolution. For both situations the answer is similar: the nonlinear nature of ReLU induces locally different effects. Thus, if we choose a pre-conditioner $A^l$ for a specific matrix $A_P^{l-1}$, it might not stabilize the matrix product for matrices $A_{P^*}^{l-1}$ corresponding to different input polytopes $P^*$.
For the case of composing only orthogonal matrices, consider a network up to layer $l-1$, where the linearization $A_P^{l-1}$ has orthogonal columns (assume the network gets larger, thus $A_P^{l-1}$ has more rows than columns). Then, the application of ReLU in form of $A^l D_{I^l} A_P^{l-1}$ removes the orthogonality property of the rows of $A_P^{l-1}$, if setting entries in the rows from $I^l$ to zero results in non-orthogonal columns (likely when considering dense matrices). Hence, $D_{I^l} A_P^{l-1}$ is not orthogonal for some $I^l$. In this case, the matrix product $A^l D_{I^l} A_P^{l-1}$ is not orthogonal, which results in decaying singular values.

This is why, even when especially designing the network by e.g. orthogonal matrices, stability issues with respect to the inverse arise. To conclude this section, we remark that the presented results are rather of a *qualitative* nature showcasing effects of ReLU on the singular values. Yet, the analysis does not require any assumptions and is thus valid for any MLP (including CNNs without pooling). To give an idea of *quantitative* effects we study numerical examples in the subsequent subsection.

## 3.2 NUMERICAL ANALYSIS

In this section, we show how the previously discussed theoretical stability properties can be examined for a given network. In particular, we conduct experiments on CIFAR10 (Krizhevsky & Hinton, 2009) using two baseline CNNs, see A4 for details on architectures and training setup. Our CNNs use only strides instead of pooling and use no residual connections and normalization layers. Thus,

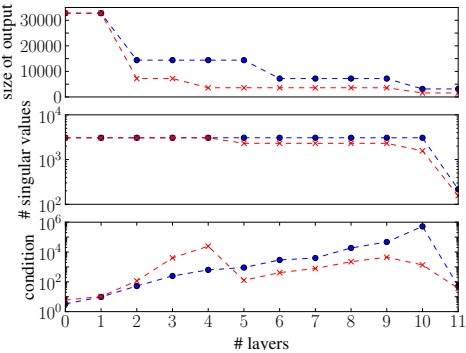

Figure 5: Blue: WideCIFAR, Red: ThinCIFAR. Top: number of output units per layer, Middle: number of singular values, Bottom: Behavior of condition number, each curve over the layers. Here, layers are split into conv-layer and ReLU-activation layer. Singular values and condition number are the median over 50 samples from the CIFAR10 test set.

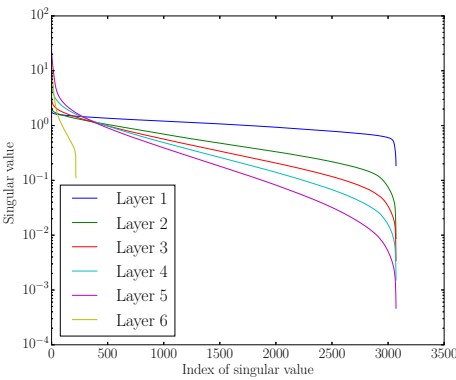

Figure 6: Decay of singular values over the layers of the network. Here, each layer includes the convolution and ReLU-activation. Reported number are taken from median over 50 samples.

the architectures fit to the theoretical study as the strided discrete convolution can be written as a matrix-vector multiplication.

**Singular values over multiple layers:** Experimentally most interesting is the development of singular values over multiple layer as several effects are potentially at interplay. Figure 6 shows how all singular values evolve in convolutional layers (layers 1-6, after application of ReLU). While the shape of the curve is similar for layer 1-5, it can be seen that the largest singular value grows, while the small singular values decrease significantly. Note that this growth of the largest singular values is in line with observations for adversarial examples, see Szegedy et al. (2014). While many defense strategies like Cisse et al. (2017) or Jia (2017) focus on the largest singular value, the behavior of the smaller singular values is often overlooked. Additionally, we provide in Appendix A5 a numerical analysis of the condition from Lemma 9 to gain an understanding of possible effects of ReLU on the singular values. Furthermore, we add results for a thinner CNN (ThinCIFAR) and for the MLP from Section 2.2 in Appendix A6.

**Relationship between stability and invariance:** While invariance is characterized by zero singular values, the condition number only takes non-zero singular values into account, see e.g. layer 6 from Figure 6. This tight relationship is further investigated in Figure 5 which compares the output size, the condition number and the number of non-zero singular values vs. the layers for WideCIFAR and ThinCIFAR. In combination with lower output dimension, ReLU has a different effect for ThinCIFAR. The number of singular values decreases in layer 5, which cuts off the smallest singular values, resulting in a lower condition number. Yet, there are more invariance directions within the corresponding linear region. For a visual comparison see Figure 1.

**Computational costs and scaling analysis:** First, we remark that the linearization of a network $F$ for an input point $x^0$ can be computed via backpropagation. Based on this linearization the computation of the full SVD scales cubically. Especially, early CNN-layers have high dimensional outputs which may cause memory issues when computing the entire SVD. We thus choose a small CNN trained on CIFAR10 as these inputs are only of size $32 \times 32 \times 3$. To scale this analysis up to e.g. ImageNet with VGG-networks, a restriction to a window of the input image is necessary to reduce the complexity of the full SVD especially for early layers. See Jacobsen et al. (2018), where the singular values restricted to input windows were used to estimate the stability of the entire i-RevNet trained on ImageNet.

## 4 SCOPE

**Characterization of preimages over multiple layers:** Theorem 4 yields a characterization (singleton, finite, infinite) of the preimage of a point $y$ under a single ReLU-layer. When considering the preimage of $y$ under multiple layers $l$ to $l - i$, two difficulties arise: 1) If the preimage of $y$ under layer $l$ is not a singleton, one needs to compute the intersection of the image of layer $l - 1$ and the preimage of $y$ under layer $l - 1$. 2) For all points in the intersection, the conditions of Theorem 4 need to be checked, which requires the solution of a linear program, see Algorithm 1 in Appendix A3. Hence, our analysis is currently restricted to a layer-by-layer approach. However, this layer-wise and local study could enable to pin down the specific layers where information, which is expressed in non-singleton preimages.

**Preimages for convolutional layers:** In general, convolutional layers are affine transforms $Ax + b$, but have a sparse and shared structure compared to dense matrices used in MLPs. Thus, the trivial case $\text{rank}(A|_{y \succ 0}) = n$ (Algorithm 1, Appendix A3) needs to be explicitly checked. For MLPs it was assumed in section 2.2 that $\text{rank}(A|_{y \succ 0}) = |\{i : y_i > 0\}|$ as dense matrix rows are almost surely linear independent in practice.

**Inverse stability for convolutional networks:** For inverse stability, we consider the linearization $A_P = A^L D_{I^{L-1}} A^{L-1} \cdots D_{I^1} A^1$ for an input polytope $P$. In convolutional networks, each $A^l$ implements a multi-channel discrete convolution. While singular values of each $A^l$ can be efficiently computed by leveraging the convolutional structure, see Sedghi et al. (2018), the shared structure is not preserved in the matrix chain $A_P$ due to the application of ReLU (expressed via $D_{I^l}$). Thus, a tighter analysis that leverages the convolutional structure in $A^l$, compared to our general assumption that $A^l$ can be any linear mapping, is not straightforward with current tools but would certainly lead to further insights.

**Extension of inverse stability across polytopes:** In our stability analysis, we employ a piecewise-linear viewpoint which allows to characterize stability via the singular values of the linearization which is exact within an input polytope. However, when considering an $\varepsilon$-ball $B_\varepsilon(y)$ around a point $y = A_P x + b_p$ to model e.g. reconstruction from noisy activations $y$, further questions arise: 1) Can all points in $B_\varepsilon(y)$ be reached by an $x^*$ from the polytope $P$? 2) Are points from another polytope $P'$ mapping to points in $B_\varepsilon(y)$? In this case, the inverse stability needs to be augmented by nonlinear considerations to model movements between piecewise-linear regions.

## 5 PRACTICAL IMPLICATIONS

While the focus of this work was an in-depth analysis of potential effects on crucial properties like invariance and robustness due to ReLU, we envision several practical implications of our approach:

**Network design and regularization:** As both the concept of omnidirectionality and removal of rows due to ReLU showed, there is a breadth of potential effects. In terms of network design, controlling such effects could be desirable. In particular, a change to injective activation functions (tanh, leakyReLU, ELU etc.) remove the discussed preimages, but immediately transfer to an instable inverse due to saturation. Furthermore, Lemma 9 draws a connection to regularizing correlation between feature maps as introduced in Rodríguez et al. (2017). Hence, both omnidirectionality and correlation between rows can be thought of as geometrical properties which could partially be controlled by regularization or architecture design. Furthermore, the analysis also shows the difficulty of controlling these properties in vanilla architectures. However, by incorporating additional structure like dimension splitting in reversible networks (Jacobsen et al., 2018) or invertible residual connections (Behrmann et al., 2018), the preimage is by design a singleton.

**Connection to information loss:** Our analysis is tightly related to mutual information $I(x^l; x)$ loss, which has gained growing interest due to the information bottleneck (Tishby & Zaslavsky, 2015; Saxe et al., 2018). In particular, invariance in layer $l$ may induce $I(x^l; x) \leq I(x^{l-1}; x)$ due to the data processing inequality (Cover & Thomas, 2006). Similarly, an instable inverse can induce an information loss as activations $x^l$ are quantized due to finite precision on hardware.

**Implications for adversarial examples:** Despite being crucial for many discriminative tasks to contract the space along uninformative directions (Mallat, 2016), invariance and robustness may induce severe vulnerabilities for adversarial examples (Szegedy et al., 2014). For instance, a model

Figure 7: Invariances of the first layer (100 ReLU neurons) of a vanilla multilayer perceptron (MLP). Despite the semantically very different examples, the features are identical as the original image "3" and the two perturbed variants "6" and "4" are in the same preimage. Further details in Appendix A7.

would be flawed if perturbations that alter the semantics only have a minor impact on the features of the network. Classically, adversarial examples are viewed as small perturbations which induce large changes in the network outputs (Goodfellow et al., 2014). Yet, reversing this perspective leads to another failure case: if large changes in the input alter its semantics for a given task, but the networks output is robust or even invariant to such changes, the model might just be as flawed from this reverse point of view.

This change in perspective leads to a natural way of addressing invariance and robustness via invertibility: If $F$ is invariant to perturbations $\Delta x$, then $x$ and $x + \Delta x$ lie in the preimage of the output $z = F(x)$ i.e. $F$ is not uniquely invertible. Robustness towards large perturbations induces an instable inverse mapping as small changes in the output can be due to large changes in the input.

Finally Figure 7 demonstrates such a striking failure, where perturbations alter the semantics drastically, yet the activations even after the first layer are identical. To find these examples, we leveraged the developed theory about preimages and a linear programming formulation, see Appendix A7.

## 6 CONCLUSION AND OUTLOOK

We presented the inverse as an approach to tackle the invariance and robustness properties of ReLU networks. Particularly, we studied two main effects: 1) conditions under which the preimage of a ReLU layer is a point, finite or infinite and 2) how ReLU can effect the inverse stability of the linearization. By deriving approaches to numerically examine these effects, we highlighted the broad range of possible effects. Moreover, controlling such properties may be desirable as our experiment on adversarial examples showed.

Besides the open questions on how to control the structure of preimages and inverse stability via architecture design or regularization, we envision several theoretical directions based on our work. Especially, incorporate nonlinear effects like moving between linear regions of rectifier networks could lift the analysis closer to practice. Furthermore, studying similarities of omnidirectionality as a geometrical property and singular values could further strengthen the link between these two crucial properties.

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

## A1  APPENDIX FOR SECTION 2

**Proof (Corollary 2, Equivalences of omnidirectionality)** We show the equivalences by proving *i)* $\Rightarrow$ *ii)* $\Rightarrow$ *iii)* $\Rightarrow$ *iv)* $\Rightarrow$ *i)*.
Let $A \in \mathbb{R}^{m \times n}$ be omnidirectional, i.e. for every $x \neq 0$, it holds that $Ax \npreceq 0$. This is equivalent to

$$Ax \preceq 0 \Rightarrow x = 0,$$

which is *ii)*. The implications from *ii)* to *iii)* and from *iii)* to *iv)* are obvious. From *iv)*, we have that

$$\forall x \neq 0 \, \exists \, i \in \{1, \ldots, m\} : \langle x, a_i \rangle > 0,$$

which is equivalent to *i)*, the omnidirectionality of $A$. Altogether, this shows the equivalence of all four points. $\qquad\square$

**Definition 10 (Convex hull)** *For $A \in \mathbb{R}^{m \times n}$, the convex hull is defined as*

$$\mathrm{Conv}(A) = \left\{ \sum_{i=1}^{m} \alpha_i a_i : \forall i \; \alpha_i \in \mathbb{R}_{\geq 0} \wedge \sum_i \alpha_i = 1 \right\},$$

*where $a_i \in \mathbb{R}^n$ are the rows of $A$.*

**Theorem 11 (Stiemke's theorem, see Dantzig (1963))** *Let $A \in \mathbb{R}^{m \times n}$ be a matrix, then the following two expressions are equivalent.*

- $\nexists y : Ay \gneq 0$

- $\exists x \succ 0 : A^T x = 0$

*Here $z \gneq 0$ means that $0 \neq z \preceq 0$ .*

**Theorem 12 (Singleton solutions of inequality systems)** *Let $A \in \mathbb{R}^{m \times n}$, $b \in \mathbb{R}^m$ and $x \in \mathbb{R}^n$. Furthermore, let the inequality system*

$$Ax + b \preceq 0,$$

*written as $(A, b)$, have a solution $x_0$.*
*Then this solution is unique if and only if there exists an index set, I, for the rows s.t. $(A|_I, b|_I)$ is omnidirectional for $x_0$.*

**Proof (Theorem 12, Singleton solutions of inequality systems)**
"$\Leftarrow$"
Let $(A|_I, b|_I)$ be omnirectional for $x_0$. Then it holds that $A|_I x + b|_I = A|_I (x - x_0) \preceq 0$. Due to the omnidirectionality of $A|_I$, $x_0$ is the unique solution of the inequality system $A|_I x + b|_I \preceq 0$. The existence of a solution for the whole system $Ax + b \preceq 0$ is guaranteed by assumption and therefore $x_0$ is the unique solution of $Ax + b \preceq 0$.
"$\Rightarrow$"
Here we will prove

"$\nexists I : (A|_I, b|_I)$ omnidirectional for some $p \Rightarrow$ solution non-unique".

We will start by doing the following logical transformations:

$$\nexists I : (A|_I, b|_I) \text{ omnidirectional for some } p$$
$$\Leftrightarrow \nexists (I, p) : (A|_I \text{ omnidirectional} \wedge b|_I = -A|_I p)$$
$$\Leftrightarrow \forall (I, p) : \neg(A|_I \text{ omnidirectional} \wedge b|_I = -A|_I p)$$
$$\Leftrightarrow \forall (I, p) : (A|_I \text{ not omnidirectional} \vee b|_I \neq -A|_I p).$$

Now we define the vector $c_0 := Ax_0 + b \preceq 0$ and the set $I$ as the index set given via $c_0 = 0$.

This means that $A|_I$ is not omnidirectional, because otherwise $A|_I x_0 + b|_I = 0$ due to the definition of $I$, which would lead to the contradiction that $(A|_I, b|_I)$ is omnidirectional for $x_0$. But this means $\exists x' \neq 0 : A|_I x' \preceq 0$ as a result of Corollary 2. Since $A|_{I^c} x_0 + b|_{I^c} \prec 0$, we also have

$\forall x \; \exists \epsilon > 0 : A|_{I^c}(x_0 + \epsilon x) + b|_{I^c} \prec 0$. This holds in particular for $x'$, so we define accordingly $x^* := \varepsilon x' \neq 0$. Therefore, we have $A|_{I^c}(x_0 + x^*) + b|_{I^c} \prec 0$ as well as

$$A|_I(x_0 + x^*) + b_I = \underbrace{A|_I x_0 + b_I}_{=c_0=0} + \underbrace{\varepsilon A|_I x'}_{\preceq 0} \preceq 0.$$

Altogether it holds that $A(x_0 + x^*) + b \preceq 0$ with $x^* \neq 0$, which means that $x_0$ is a non-unique solution for the inequality system $Ax + b \preceq 0$. $\square$

**Proof (Theorem 4, Preimages of ReLU-layers)** We consider the ReLU-layer

$$y = \text{ReLU}(Ax + b),$$

given its output $y \in \mathbb{R}^m$ with $A \in \mathbb{R}^{m \times n}$, $b \in \mathbb{R}^m$ and $x \in \mathbb{R}^n$. Clearly, this equation can also be written as the mixed linear system

$$A|_{y \succ 0} x + b|_{y \succ 0} = y|_{y \succ 0},$$
$$A|_{y=0} x + b|_{y=0} \preceq 0.$$

This allows us to consider the two cases

$$\mathcal{N}(A|_{y \succ 0}) = \{0\} \text{ and } \mathcal{N}(A|_{y \succ 0}) \neq \{0\}.$$

In the first case, we have a linear system which allows us to calculate $x$ uniquely, i.e. we can do retrieval. This leads us to the second case, the interesting one. In this case we can only recover $x$ uniquely if and only if the system of inequalities "pins down" $P_{\mathcal{N}(A|_{y \succ 0})} x$, where $P_V$ is the orthogonal projection into the closed space $V$. Formally this requires

$$A|_{y \preceq 0}(P_{\mathcal{N}(A|_{y \succ 0})^\perp} x + P_{\mathcal{N}(A|_{y \succ 0})} x) + b|_{y \preceq 0} \preceq 0$$

to have a unique solution for $x \in \mathbb{R}^n$ and $P_{\mathcal{N}(A|_{y \succ 0})^\perp} x$ fixed (given via the equality system). By defining $\bar{b} := b|_{y \preceq 0} + A|_{y \preceq 0}(P_{\mathcal{N}(A|_{y \succ 0})^\perp} x)$ we have

$$A|_{y \preceq 0}(P_{\mathcal{N}(A|_{y \succ 0})} x) + \bar{b} \preceq 0.$$

If $O \in \mathbb{R}^{k \times n}$ now denotes an orthonormal basis of $\mathcal{N}(A|_{y \succ 0})$, where $k := \dim \mathcal{N}(A|_{y \succ 0})$, we can write

$$\overline{A}\overline{x} + \bar{b} \preceq 0,$$

where $\overline{A} := AO^T$ and $\overline{x} := Ox$ is a general element in $\mathbb{R}^k$. It now follows from Theorem 12 that the inequality system $(\overline{A}, \bar{b})$ has a unique solution if and only if $(\overline{A}, \bar{b})$ has a subset of rows that are omnidirectional for some point $p$. $\square$

**Proof (Theorem 5, Convex hull)** Since $\mathcal{N}(A) = \{0\}$ follows from both sides of the equivalence, the following sequence of equivalencies holds. $0 \in \text{Conv}(A)^\circ \Leftrightarrow \exists x \succ 0 : A^T x = 0 \overset{\text{Theorem 11}}{\Longleftrightarrow} \nexists y : Ay \gneqq 0$. Together with $\mathcal{N}(A) = \{0\}$, which means that $\nexists y \neq 0 : Ay = 0$, leads altogether to $\nexists y \neq 0 : Ay \preceq 0$. $\square$

## A2  PROOFS FOR SECTION 3

**Proof (Lemma 8, Global upper bound for largest and smallest singular value)** The upper bound on the largest singular value is trivial, as ReLU is contractive or in other terms $\|D_I Ax\|_2 \leq \|Ax\|_2$ for all $I$ and $x \in \mathbb{R}^n$.

To prove the upper bound for the smallest singular value, we assume

$$\sigma_M := \min\{\sigma_l : \sigma_l > 0\} > \tilde{\sigma}_k \tag{4}$$

and aim to produce a contradiction. Consider all singular vectors $\tilde{v}_{k^*}$ with $k^* \geq k$ from matrix $A$. It holds for all $\tilde{v}_{k^*}$

$$\tilde{\sigma}_k \geq \tilde{\sigma}_{k^*} = \|A\tilde{v}_{k^*}\|_2 \geq \|D_I A\tilde{v}_{k^*}\|_2, \tag{5}$$

as $D_I$ is a projection matrix and thus only contracting. As

$$\sigma_M = \min_{\substack{\|x\|_2=1 \\ x \in \mathcal{N}(D_I A)^\perp}} \|D_I Ax\|_2,$$

all $\tilde{v}_{k^*} \notin \mathcal{N}(D_I A)^\perp$. Otherwise, a $\tilde{v}_{k^*}$ would be a minimizer by estimation 5, which would violate the assumption 4.

Due to $\mathcal{N}(D_I A)^\perp \oplus \mathcal{N}(D_I A) = \mathbb{R}^n$, it holds $\tilde{v}_{k^*} \in \mathcal{N}(D_I A)$. As $\tilde{v}_{k^*}$ are orthogonal, $\dim(span(v_{k^*})) = |I| + 1$ (note: $k^* = k, ..., N$ and $k = N - |I|$, thus there are $|I| + 1$ singular vectors $v_{k^*}$ in total). Furthermore, $\tilde{v}_{k^*}$ were not in $\mathcal{N}(A)$ by definition (corresponding singular values were strictly positive).

Hence, the nullspace of $D_I$ must have $\dim(\mathcal{N}(D_I)) \geq |I| + 1$. But $D_I$ is the identity matrix except $|I|$ zeros on the diagonal, thus $\dim(\mathcal{N}(D_I)) = |I|$, which yields a contradiction. $\qquad\square$

**Proof (Lemma 9, Removal of weakly correlated rows)** Consider $v = \frac{a_k}{\|a_k\|_2}$. Then,

$$(D_I A v)_k = 0,$$

since $k \in I$ ($k$-th row of $D_I$ is zero). Furthermore, for all $j \neq k$ it holds by condition 3

$$(D_I A v)_j = \frac{\langle a_k, a_j \rangle}{\|a_k\|_2} \leq \frac{|\langle a_k, a_j \rangle|}{\|a_k\|_2} \leq \frac{c}{\sqrt{M}}.$$

Hence,

$$\|D_I A v\|_2 = \sqrt{\sum_{j \notin I} \left( \frac{\langle a_k, a_j \rangle}{\|a_k\|_2} \right)^2} \leq \sqrt{M \left( \frac{c}{\sqrt{M}} \right)^2} = c.$$

As $a_k \in \mathcal{N}(D_I A)^\perp$, $v \in \mathcal{N}(D_I A)^\perp$ as well. Thus,

$$\sigma_K = \min_{\substack{\|x\|_2 \\ x \in \mathcal{N}(D_I A)^\perp}} \|D_I A\|_2 \leq \|D_I A v\|_2 \leq c.$$

$\qquad\square$

## A3    APPENDIX FOR SECTION 2.2

In this section, we formulate the algorithm to determine whether the preimage of $y$ given by

$$y = \text{ReLU}(Ax + b)$$

is finite.
This requires to check whether $\overline{A}$ (see Theorem 4) is omnidirectional, which is equivalent to

$$0 \in \text{Conv}(\overline{A})^{\text{o}},$$

see Theorem 5. Since it is reasonable to assume that $0$ will not lie on the boundary of the convex hull, we can formulate this as a *linear programming* problem. The side-conditions incorporate the definition of convex hulls (Definition 10, Appendix A1). The objective function is chosen arbitrary, as we are only interested in a solution.

---

**Algorithm 1** Finite preimage

---

    **Input:** $A \in \mathbb{R}^{m \times n}$, $b \in \mathbb{R}^m$, $y \in \mathbb{R}^m$
    **if** $\text{rank}(A|_{y \succ 0}) = n$ **then**
        **return** True {Preimage is a singleton}
    **end if**
    $O \leftarrow$ orthonormal basis of $\mathcal{N}(A|_{y \succ 0})$, ($\in \mathbb{R}^{k \times n}$)
    $\overline{A} \leftarrow A|_{y=0}O^T$, ($\in \mathbb{R}^{\tilde{k} \times k}$)
    **if** $\tilde{k} \leq k$ **then**
        **return** False {see Corollary 6}
    **end if**
    $c \leftarrow (1; \ldots; 1)$ {arbitrary objective}

    **return** Does a solution for the linear program $\begin{cases} \max c^T x \\ \text{subject to} \\ \quad\quad \overline{A}^T x = 0 \\ \quad\quad (1; \ldots; 1)^T x = 1 \\ \quad\quad x \in [0, 1]^{\tilde{k}} \end{cases}$ exists?

---

## A4    ARCHITECTURES FOR NUMERICAL STUDIES

Training details for MLP on MNIST:

- Training using Adam optimizer (Kingma & Ba, 2015)
- Epochs: 25
- Batch size: 1000

Training details for WideCIFAR and ThinCIFAR:

- Training setup from Keras (Chollet et al., 2015) examples: `cifar10_cnn`
- No data augmentation
- RMSprop optimizer
- Epochs: 100
- Batch size: 32

Table 1: Architecture of MLP trained on MNIST

| Index | Type | kernel size | stride | # feature maps | # output units |
|---|---|---|---|---|---|
| 0 | Input layer | - | - | 3 | |
| 1 | Dense layer | - | - | - | 100 |
| 2 | Dense layer | - | - | - | 100 |
| 3 | Dense layer | - | - | - | 100 |
| 4 | Dense layer | - | - | - | 100 |
| 5 | Dense layer | - | - | - | 100 |
| 6 | Dense layer | - | - | - | 100 |
| 7 | Dense layer | - | - | - | 100 |
| 8 | Dense layer | - | - | - | 100 |
| 9 | Dense layer | - | - | - | 100 |
| 10 | Dense layer | - | - | - | 100 |
| 11 | Dense layer (softmax) | - | - | - | 10 |

Table 2: Architecture of MLP trained on MNIST

| Index | Type | kernel size | stride | # feature maps | # output units |
|---|---|---|---|---|---|
| 0 | Input layer | - | - | 3 | |
| 1 | Dense layer | - | - | - | 3500 |
| 2 | Dense layer | - | - | - | 784 |
| 3 | Dense layer (softmax) | - | - | - | 10 |

Table 3: Architecture of WideCIFAR

| Index | Type | kernel size | stride | # feature maps | # output units |
|---|---|---|---|---|---|
| 0 | Input layer | - | - | 3 | |
| 1 | Convolutional layer | (3,3) | (1,1) | 32 | - |
| 2 | Convolutional layer | (3,3) | (2,2) | 64 | - |
| 3 | Convolutional layer | (3,3) | (1,1) | 64 | - |
| 4 | Convolutional layer | (3,3) | (1,1) | 32 | - |
| 5 | Convolutional layer | (3,3) | (1,1) | 32 | - |
| 6 | Convolutional layer | (3,3) | (2,2) | 64 | - |
| 7 | Dense layer | - | - | - | 512 |
| 8 | Dense layer (softmax) | - | - | - | 10 |

Table 4: Architecture of ThinCIFAR

| Index | Type | kernel size | stride | # feature maps | # output units |
|---|---|---|---|---|---|
| 0 | Input layer | - | - | 3 | |
| 1 | Convolutional layer | (3,3) | (1,1) | 32 | - |
| 2 | Convolutional layer | (3,3) | (2,2) | 32 | - |
| 3 | Convolutional layer | (3,3) | (1,1) | 16 | - |
| 4 | Convolutional layer | (3,3) | (1,1) | 16 | - |
| 5 | Convolutional layer | (3,3) | (1,1) | 16 | - |
| 6 | Convolutional layer | (3,3) | (2,2) | 32 | - |
| 7 | Dense layer | - | - | - | 512 |
| 8 | Dense layer (softmax) | - | - | - | 10 |

## A5 EFFECT OF RELU NUMERICAL ANALYSIS OF LEMMA 9

In order to better understand the bound on the smallest singular value after ReLU, given by Lemma 9, we numerically proceed as follows:

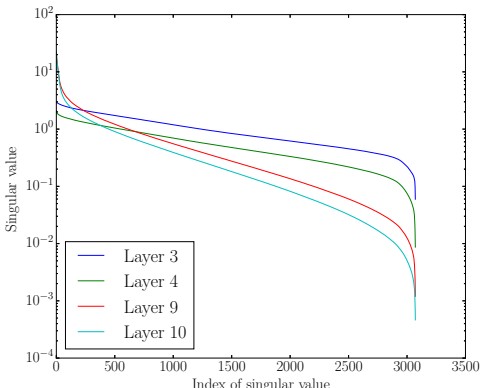

Figure 8: Effect of ReLU on the singular values for WideCifar. The curves show the effect in layer 2 (layer 3 and 4 in the legend, because ReLU is counted as an extra activation layer) and layer 5 (layer 9 and 10), where each curve is the median over 50 samples.

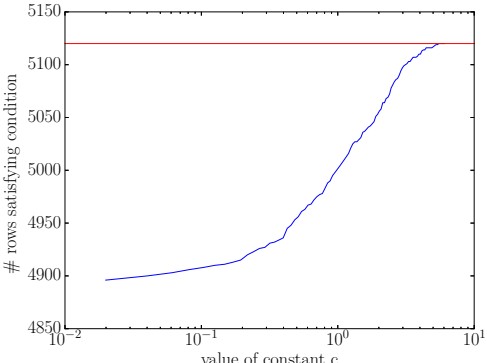

Figure 9: Curve showing how many rows $a_i$ satisfy condition 3 from Lemma 9 depending on values of constant $c$. The red line shows the total number of remaining rows after removal by ReLU, $M = 5120$. Even for small constants $c$ most $a_i$ fulfill condition 3, yet not all, which is required by the lemma to give an upper bound on the smallest singular value. The example is from layer 4 of WideCIFAR, for only one sample from the test set.

1. We choose $c \in [a, b]$, where $a, b$ are suitable interval endpoints.

2. Given $c$, we compute for every $a_k$ with $k \in I$ the value of $c\frac{\|a_k\|_2}{\sqrt{M}}$ ($M$ is the number of remaining rows, in the example $M = 5120$).

3. For every $a_k$ we count the number of $a_i$ satisfying

$$|\langle a_i, a_k \rangle| \le c\frac{\|a_k\|_2}{\sqrt{M}}.$$

4. We take the $a_k$ with the maximal number of $a_i$ satisfying the condition. (Note, that this ignores the requirement $a_k \in \mathcal{N}(D_I A)^\perp$.)

5. If we have an $a_k$, where all $a_i$ satisfy the condition, the corresponding constant $c$ gives the upper bound on the smallest singular value after ReLU.

Figure 9 shows the number of $a_i$ satisfying the correlation condition given different choices of $c$. The red line is reached for $c \approx 6$. However, even the largest singular value after ReLU is smaller than 2.5 (shown in Figure 8). Thus, the bound given by Lemma 9 is far off. This can be explained by the

fact, that this situation is quite redundant ($M = 5120$) and there are rows $a_i$ still correlated to the removed rows $a_k$.

However, in the further Experiments on ThinCIFAR, we observe (see Figure A6) a stronger effect of $\mathrm{ReLU}$ in layer 2, which can be explained by having a less redundant scenario with fewer remaining rows.

## A6  FURTHER EXPERIMENTS

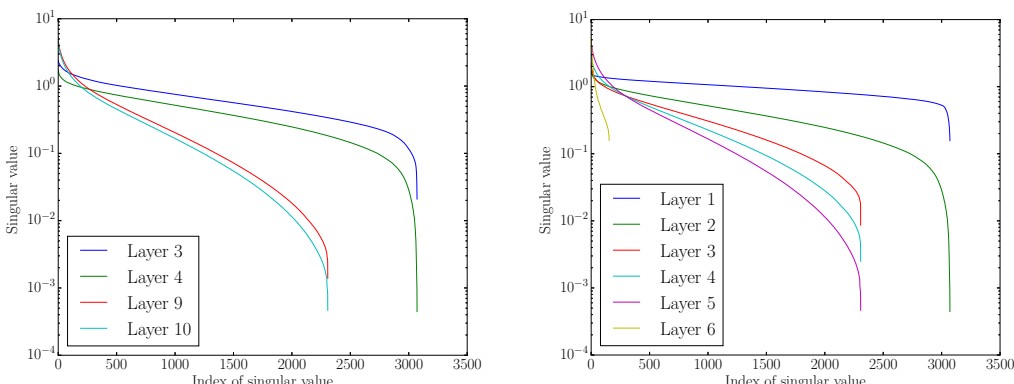

Figure 10: Left: Effect of ReLU on the singular values for ThinCifar. The curves show the effect in layer 2 (layer 3 and 4 in legend, because ReLU is counted as an extra activation layer) and layer 5 (layer 9 and 10). Right: Decay of singular values over the layers ThinCifar. Here, each layer includes the convolution and ReLU-activation. Reported number are taken from median over 50 samples. Best viewed in color.

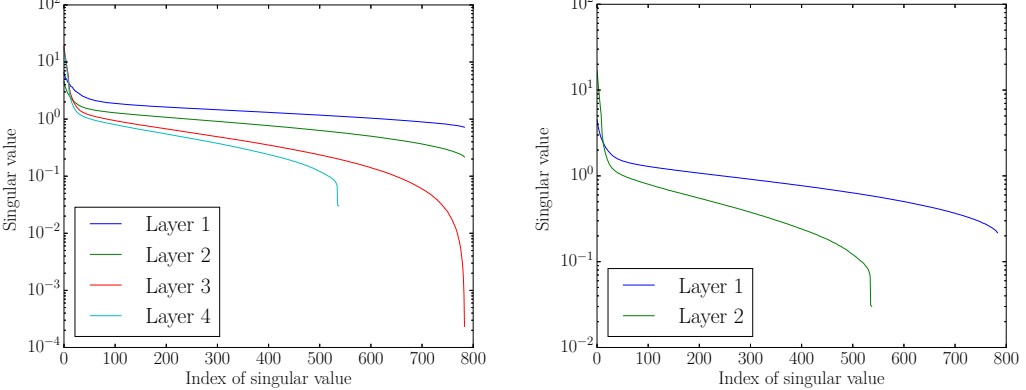

Figure 11: Left: Effect of ReLU on the singular values for the MLP on MNIST. The curves show the effect in layer 1(layer 1 and 2 in legend, because ReLU is counted as an extra activation layer) and layer 2 (layer 3 and 4). Right: Decay of singular values over the layers of MLP on MNIST. Here, each layer includes the fully-connected layer and ReLU-activation. Reported number are taken from median over 50 samples.

## A7  INVARIANCE EXPERIMENT USING AN MLP ON MNIST

This section briefly describes how the results in Figure 7 from the introduction were obtained (copied in Figure 12 for readability). After training the network from 1 (in Appendix A4), we searched the MNIST test set for input images with yielded the fewest positive activations in the first layer, in the figure the digits "3" and "4". After selecting the example input $x^*$, we selected another input $c$ belonging to a different class (e.g. a "6" and "4" in the first example).

Figure 12: Invariances of the first layer (100 ReLU neurons) of a vanilla MLP. (Exact architecture in Appendix A4 Table 1.)

Afterwards, we solved following linear programming problem to find a perturbed $x$:

$$\begin{cases} \max \; \langle c, x \rangle \\ \text{subject to} \\ \quad A|_{y^* \succ 0} x + b|_{y^* \succ 0} = y^*|_{y^* \succ 0} \\ \quad A|_{y^* \prec 0} x + b|_{y^* \prec 0} \preceq 0 \\ \quad x \in [0,1]^{\tilde{k}} \end{cases} ,$$

where the features of the first layer are computed via

$$y^* = \text{ReLU}(Ax^* + b).$$

Hence, we searched within the preimage of the features $y^*$ of the first layer for examples $x$ which resemble images $c$ from another class. By doing this we observe, that the preimages of the MLP may have large volume. In these cases, the network is invariant to some semantics changes which shows how the study of preimages can reveal previously unknown properties.

