# OpenReview forum: "Invariance and Inverse Stability under ReLU"
_ICLR.cc/2019/Conference_

### Official Review · AnonReviewer2 · 2018-11-02
**interesting and original idea, not sure about practical implications**

**Rating:** 7
**Confidence:** 3

**Review:**

This paper presents an analysis of the inverse invariance of ReLU networks. It makes the observation that one can describe the pre-image of an image point z = F(x) using linear algebra arguments. They provide necessary conditions for the pre-image to be a singleton or a finite volume polytope. They also provide upper-bounds on the singular values of a train network and measure those in standard CNNs.

The paper is well-written but the structure is a bit disconnected; most notably, I didn't see clearly how Section 2 and 3 fit together. The proofs seem correct and rely mostly on elementary linear algebra argument; this simplicity makes the analysis quite interesting. The argument about a different kind of adversarial examples is also very interesting; instead of looking for small perturbation that affect the mapping in drastic ways, find large perturbations that in invariant directions of the network. However, the experiments are overall not very useful to the comprehension of the paper and not that illustrative.

I have several questions for the authors:
- the conditions presented in Theorem 4, seem hard to check in practice; what is the time complexity of this operation? I believe that checking if A is omnidirectional is equivalent to an LP but how do you solve the combinatorial size of doing that over all set of indices?
- I understand the upper bounds on the singular values, but I am not sure how they relate to inverse stability. Maybe more explanation and quantitative analysis (e.g. relating the volume of the preimage of an epsilon ball around z to the singular values) could be helpful.
- Is there actionable consequences one could draw from your papers? The way the results are presented seem like they are only useful inspection after training; are your results able to derive methods to enforce conditions on the pre-images for example?

In conclusion, this paper does an interesting and original analysis which can help us understand better the polytopes composing the input space. The experiments are not very convincing or illustrative of the theoretical results in my opinion. It is not clear how those observations can affect practical algorithms and this is something I hope the author can address.

---

> ### Author Response · Authors · 2018-11-13
> **Added discussion of raised points in revision (Scope and Practical Implications)**
>
> We thank you for acknowledging the novelty our findings and your appreciation for the elementary nature of our theory.
>
> ----------------
> - Q: How do Section 2 & 3 fit together?
> Although it is true that our paper can roughly be divided into two section, we want to stress that these sections are inextricably linked due to the nature of their topics, since we see invariance as a limit case of inverse stability. We therefore think it is natural to study both of them.
> However, the analysis of the limit case, invariance, admits more powerful tools (see e.g. Theorem 4), since one is only interested in whether a singular value is zero or not. Hence, the invariance is qualitative, whereas for stability we need to quantify singular values.
>
> -----------------
> -Q: Combinatorial problem to check Theorem 4:
> While there are indeed a combinatorial number of possible tuples that the Theorem 4 describes, we can use the following trick in the design of the Algorithm 1 (Appendix A3) to circumvent these computations: The set of tuples (A, b) that form omnidirectional tuples is a null-set in all tuples of same form, we therefore ignore this case in our numerical analysis. Hence, we only have to check whether we have a compact or unbounded preimage. This can be done by simply checking whether A is omnidirectional or not.
>
> ----------------
> -Q: Upper bounds and inverse stability:
> The smallest singular values are directly linked to inverse stability for points from the same input polytope (where the linearization is exact). The upper bounds (Lemma 9) and the correlation effect are interesting, as they show how a well-conditioned matrix (subset of rows almost orthogonal) may become instable due to the removal of rows via ReLU. If the correlation of some rows is arbitrarily small (but non-zero) between remaining and removed rows, the upper bounds can be arbitrarily small. Thus, this Lemma provides an intuition how hard it is to globally control inverse stability with a vanilla architecture (linear mapping followed by ReLU).
>
> However, when considering an epsilon ball around activations, two main questions arise: 1) Are all points in the ball reachable from the considered input polytope? 2) Do points from other input polytopes map to the epsilon ball? If the second case holds, one would need to consider different linearizations of the network and thus extend the analysis to movements between the polytopes.
> -> Added a comment in the newly written “Scope” section in the revision
>
>
> -------------------
> -Q: Actionable consequences from paper:
> One consequence of our paper is that it is close to impossible (each layer need at least to double the number of neuron) to enforce invertibility and it is similarly hard to enforce compactness in ReLU layers. This leads to the conclusion that if one wants invertibility or even just compactness reliably over the whole space, vanilla architectures using ReLU are not a good tool for the task.
> Hence, our analysis can be seen as an argument for additional structure like dimension splitting in reversible networks (see e.g. Jacobsen et al. (2018)). These structures allow for guarantees as they are by design bijective, while vanilla architectures show a breadth of possible effects as shown in our analysis.
> -> Added a comment to “Practical Implications” in the revision
>
> - Q: Illustrative experiments:
> We currently thinking about an experiment to better illustrate the intuition of our theory and would appreciate any suggestions.
>
> We thank the reviewer for the helpful comments and we would appreciate further suggestions.

---

> > ### Author Response · Authors · 2018-11-20
> > **Added illustration**
> >
> > We added an illustrative example in the introduction to give an intuitive understanding of invariance, stability and their relationship.
> > We would appreciate further suggestions.

---

> > > ### Comment · AnonReviewer2 · 2018-11-26
> > > **Response**
> > >
> > > I thank the authors for their clarifications and appreciate that they have addressed some of my concerns in the paper. As a results of the response, my rating remains unchanged.

---

### Official Review · AnonReviewer3 · 2018-11-03
**not entirely novel with few concerns but includes results leading to interesting insights**

**Rating:** 6
**Confidence:** 4

**Review:**

The paper has two distinct parts. In the first part (section 2) it studies the volume of preimage of a ReLU network’s activation at a certain layer as being singular, finite, or infinite. This part is an extension of the work in the study of (Carlsson et al. 2017). The second part (section 3) builds on the piecewise linearity of a ReLU network’s forward function. As a result, each point in the input space is in a polytope where the model acts linearly. In that respect, it studies the stability of the linearized model at a point in the input space. The study involves looking at the singular values of the linear mapping.

The findings of the paper are non-trivial and the implications potentially interesting. However, I have some concerns about the study.

There is a key concern about the feasibility of the numerical analysis for the first part. That is, a layer-by-layer study can have a computational problem where the preimage is finite at each layer but can become infinite by the mapping of the preceding layers. In that regard, I would like the authors to comment on the worst-case computational complexity of the numerical analysis for determining the volume of a preimage through multiple layers.

As for the second part, the authors mention the increase in the dimensionality of the latent space in the current deep networks. However, this observation views convolutional networks as MLPs. However, there is more structure in a convolutional layer’s mapping function. The structure is obtained by the shared and sparse rows of matrix A. I would like the authors to comment on how the studies will be affected by this property of the common networks.

All in all, while there are some concerns and the contributions are not entirely novel, the reviewer believes the findings of the paper is generally non-trivial and shed more light on the inner workings of the ReLU networks and is thus a valuable contribution to the field.

---

> ### Author Response · Authors · 2018-11-13
> **Added discussion on raised points in revision (Scope section)**
>
> We thank you for acknowledging our findings to be useful to shed more light on the inner workings of ReLU-networks.
> We respond to your raised points below:
>
> ---------
> - Q: Algorithm applied layer-by-layer:
> As correctly observed, the application of our algorithm to classify the preimage of one data point of one ReLU layer does not easily translate to more than one layer. On the one hand, as pointed out, as soon as the preimage is no longer only a point itself it is no longer applicable. On the other hand it is a first step towards a multilayer analysis and allows a localized layer-by-layer analysis for the first time.
> -> For more on this we refer to the newly added Section “Scope” in the revision.
>
> --------
> - Q: Applicable to CNNs:
> It is true that our analysis is quite general considering MLPs and not specifically CNNs and indeed we find it very likely that there are stronger results possible for CNNs than the ones we presented.
> -> Added a discussion on CNNs in the new “Scope” Section in the revision
> ------------
> - Q: Relation to Carlsson et al. (2017):
> While the work of Carlsson et al. (2017) rather focus on a general analysis on the shape of preimages of activities at arbitrary levels and gives a first geometrical view as a piecewise linear manifold, we present in our work an in-depth understanding for preimages and the inverse mapping of ReLU networks:
> 1) We perform a qualitative analysis for the preimages and give computable conditions when the inverse image of an output is finite, infinite or a single point by performing an intuitive mathematical derivation.
> 2) We analyze the stability of the inverse mapping by investigating the singular values of the linearization of the network and confirm our theoretical results by numerical experiments.
>
> ---------
> We therefore think that our work can be seen as a significantly different approach to the one presented by Carlsson et al. (2017).
>
> We thank the reviewer for the helpful comments and would appreciate further discussions.

---

> > ### Comment · AnonReviewer3 · 2018-11-27
> > **Thanks for the comments**
> >
> > The fact that the empirical study of a multi-layer network can be prohibitively expensive diminishes the value of the first part of the work in light of Carlsson et al. (2017). Although, the view is slightly different through the proposal of the omnidirectionality. The second part of the paper is still interesting but I am not sure if it's enough contribution to warrant a publication at ICLR. So, all in all, my rating stays at borderline.

---

### Official Review · AnonReviewer4 · 2018-11-18
**Interesting investigation but needs work**

**Rating:** 6
**Confidence:** 3

**Review:**





Review

This paper discusses invariances in ReLU networks. The discussion is anchored around the observation that while the spectral norm of neural networks layers (their product bounds the Lipschitz constant) has been investigated as a measure of robustness of nets, the smallest singular values are also of interest as these indicate directions of invariance.

The paper consists mostly of a theoretical analysis with little empirical support, focusing on a property of matrices called omnidirectIonality. The definition given seems weird — an A \in R^{m \times n} is omnidirectional if there exists a unique x \in R^n such that Ax \leq 0.

If there is a *unique* x then that x must be 0. Else if there were a nonzero x for which Ax \leq 0, then A(cx) also \leq 0 for any positive scalar 0 and thus x is not unique. Moreover if x must be equal to 0 Ax \leq 0 and at that point Ax = 0, then that means there exists no x for which Ax < 0, so why not just say this outright? Perhaps a cleaner definition would just be “A is full rank and there does not exist any X such that Ax < 0? Also perhaps better to use the curly sign for vector inequality.

Overall the paper, while interesting is unacceptably messy.
The first two pages have no paragraph breaks!!! This means either that the author are separating paragraphs with \\ \noindent or that they have modified the style file to remove paragraph breaks to save space. Either choice is unreadable and unacceptable. The paper is also littered with typos and vague statements (many enumerated below under *small issues*). In this case, they add up to make a big issue.


The notation at the top of page 4 — see (1) and (2) — comes out of nowhere and requires explanation. |_{y>0} x + b |_{y>0}  <— what is the purpose of the subscripts here? Why is this notation never introduced?

Ultimately this paper focuses on the question on whether the pre-image of a ReLU layer can be concluded (based on the post-image) to be a singleton,  a compact polytope, or if it has infinite volume. The paper offers some analysis, suggesting when each of the conditions occurs, upper bounds the smallest singular value of D A (where the example dependent diagonal matrix D incorporates the ReLU activation (shouldn’t this be more clearly introduced and notated?).

Ultimately this paper is interesting but falls well below the standards of exposition that I expect from a theory paper and doesn’t go very far at connecting the analysis back to the claimed motivation of investigating practical invariances. If the authors significantly improve the quality of the draft, I’ll be happy to revisit it and re-evaluate my score.


Small issues

The following is a *very* incomplete list of small bugs found in the paper:

“From a high-level perspective both of these approaches” --> missing comma after “perspective”

"as well as the gradient correspond to the highest
possible responds for a given perturbation" --> incomprehensible "corresponding?" "possible responds?" do you mean "response", and if so what is the precise technical meaning here?

"analyzing the lowest possible response" what does "response' mean here?

"We provide upper bounds on the smallest singular value" -- the singular value of what? This hasn't been stated yet.

"reverse view on adversarial examples" --- what this means isn't clear from the preceding text.

"we aim to theoretically derive means to uncover mechanisms of rectifier networks without assumptions on the weights" -- what does "mechanisms" mean here?

Notation section -- need a sentence here at the beginning, can't just have a section heading followed by bullets.

"realated"

---

> ### Author Response · Authors · 2018-11-20
> **Improved clarity of the draft**
>
> We thank you for interest in our work and your thorough review. We found your review particularly helpful in our efforts to create a more structured and formally sound version of the paper.
>
> ------------
> - On Notation:
> + We now have an introductory sentence to our notation section to improve the flow of reading and in order to not open the section with bullet points right away.
> + As you suggested we changed our inequality notation to curly brackets to make it visually clearer that we are dealing with vectors.
> + In fact we did introduce our subscript notation of the kind "b |_{y>0}". It is defined in our section on notation. Nevertheless your troubles compelled us to restate the meaning of this notation at the time of its first usage.
>
> ------------
> - On Omnidirectionality:
> We are glad that you seem to find the concept of omnidirectionality intriguing. In the new version of our paper we therefore tried hard to make the definition as intelligible and intuitive as possible.
>
> We now use the following, equivalent, formulation as our definition: The matrix A of the form m x n is omnidirectional if for every given x in R^n \ {0} there exists a row A_i of A such that <A_i, x> > 0. Or in less formal terms: There is no open linear half-space in R^n that does not contain an A_i.
>
> This geometric formulation of the definition is not only the origin of the naming, but it is also a mathematically sound formulation similar to the one you suggested “A is full rank and there does not exist any X such that Ax < 0”. The problem with your formulation (from the point of view of our notation) lies in the usage of the inequality sign since we defined it in the notation section to be element-wise. Your formulation would therefore require every entry of Ax to be negative, while for omnidirectionality one entry would be sufficient as long as the others are non-positive.
> This miscommunication also encouraged us to change our signs to the curly version as suggested by you.
> The original reason we used the previous definition was that we thought it would show more clearly what the core property is, namely that omnidirectionality can be used to nail down one precise solution. But we are now convinced that the best introductory formulation is the geometric one, as it offers an intuition of omnidirectionality.
>
> ------------
> - On Orderliness in General:
> + As you suggested we carefully restructured our paragraphs and removed the appearances of “\\”.
> + We corrected as many typos as we could find, we would be very thankful for pointing out any further typos!
> + We tried to improve the readability by increasing structure of longer segments of text e.g. by introducing informal titles.
> + We added a clear and formal definition of the “binary” diagonal matrices representing the application of ReLU. (Section 3.1)
> + We rewrote potentially ambiguous statements in order to remove any inaccuracies.
>
>
> We hope we addressed your main concerns and our changes based on your review led to a paper that conforms with your standards of exposition. We want to thank you again for your thoughtful review and would welcome further advice.

---

> > ### Comment · AnonReviewer4 · 2018-12-05
> > **Rebuttal received**
> >
> > I am glad to see that the authors have fixed a number of the issues discussed in my review. Per these improvements in the draft, I am increasing my score from 5 to 6.

---

### Author Response · Authors · 2018-11-20
**Revision summary**

We have addressed the comments of the reviewers and updated our manuscript with an illustrative example and discussions accordingly. We are glad to see the positive comments on our work and are confident that the revised version substantially increases the quality of our manuscript.

In the following, we list our main updates:

1.) We added a new visualization in the introduction, which illustrates our theoretical work and clarifies the connection between our analysis of preimages and stability in Section 2 and 3 (see Reviewer #2).
2.) We changed the definition of omnidirectionality to another equivalent formulation, which is based on the more intuitive and intelligible geometric interpretation of this property. The equivalent formulations in Corollary 2 and its proof were changed accordingly. (see Reviewer #4)
3.) We added a new Section “Scope” to the document, which addresses several raised comments of our reviewers.
   + The characterization of preimages over multiple layers (see Reviewer #3)
   + The applicability of our work to CNN’s (see Reviewer #2)
   + The extension of the inverse stability across polytopes (see Reviewer #2)
4.) We corrected several typos and restructured the document to improve the readability and consistency of our draft (see Reviewer #4).

---

### Meta-Review · Area_Chair1 · 2018-12-17

**Confidence:** 5
**Recommendation:** Reject

**Metareview:**

The main strength of the paper is to provide a clear mathematical characterization of invertible neural networks. The reviewers and the AC also note potential weakness including 1) the exposition of the paper can be much improved; 2) it's unclear how these analyses can help improve the training algorithm or architecture design since these characterizations are likely not computable; 3) the novelty compared to previous work Carlsson et al. 2017 may not be enough for ICLR acceptance. These weakness are considered critical issues by the AC in the decision.